# Vaccination for Respiratory Syncytial Virus: A Narrative Review and Primer for Clinicians

**DOI:** 10.3390/vaccines11121809

**Published:** 2023-12-02

**Authors:** Kay Choong See

**Affiliations:** Division of Respiratory and Critical Care Medicine, Department of Medicine, National University Hospital, Singapore 119228, Singapore; kaychoongsee@nus.edu.sg

**Keywords:** adjuvants, immunologic, mRNA vaccines, pneumonia, viral, vaccination hesitancy, viral fusion proteins

## Abstract

Respiratory syncytial virus (RSV) poses a significant burden on public health, causing lower respiratory tract infections in infants, young children, older adults, and immunocompromised individuals. Recent development and licensure of effective RSV vaccines provide a promising approach to lessening the associated morbidity and mortality of severe infections. This narrative review aims to empower clinicians with the necessary knowledge to make informed decisions regarding RSV vaccination, focusing on the prevention and control of RSV infections, especially among vulnerable populations. The paper explores the available RSV vaccines and existing evidence regarding their efficacy and safety in diverse populations. Synthesizing this information for clinicians can help the latter understand the benefits and considerations associated with RSV vaccination, contributing to improved patient care and public health outcomes.

## 1. Introduction

Respiratory syncytial virus (RSV) was first called chimpanzee coryza agent in 1956, following its discovery during an outbreak investigation into a group of chimpanzees with cold-like symptoms [1]. It was then renamed as RSV after being isolated from human infants with acute respiratory disease [2,3]. Since Beem’s seminal 1960 report on RSV in children [4], RSV is now known to affect humans of all ages. RSV is the leading cause of severe lower respiratory tract disease (including pneumonia), hospitalizations, and fatalities among infants and young children [5]. Among adults (particularly in individuals aged 65 and above), RSV is one of the main causes of community-acquired pneumonia and is a major contributor to morbidity, mortality [6], rehospitalization [7], and reduced quality of life [8].

RSV is an enveloped RNA virus belonging to the *Pneumoviridae* family and is classified into two subtypes, A and B [9]. It consists of a nucleocapsid protein complex, a matrix protein layer, and three transmembrane glycoproteins. The fusion protein (F) and attachment glycoproteins (G) play crucial roles in viral entry and membrane fusion. RSV F is initially synthesized as an inactive precursor (F0) that undergoes cleavage to form fusion-competent subunits (F1 and F2). The fusion process involves conformational changes in the prefusion state of RSV F (i.e., preF), leading to the insertion of the fusion peptide into the target membrane and the formation of a six-helix bundle that facilitates membrane fusion. Currently available vaccines [10,11] target the highly conserved preF and aim to generate neutralizing antibodies against it.

RSV infections commonly result in seasonal outbreaks worldwide [12]. In the northern hemisphere, these outbreaks typically happen between October to May. Conversely, in the southern hemisphere, outbreaks usually occur from May to September. In tropical and semitropical regions, seasonal outbreaks of RSV are less distinct compared to temperate regions and are often associated with the rainy seasons and humid conditions [13,14].

Nearly all children experience RSV infection by two years of age, and reinfections are frequent. RSV is the main cause of lower respiratory infections in children [5], which, in turn, is the second most common cause of mortality in children younger than five years in 2019 [15]. Globally, RSV is responsible for one out of every 50 deaths in children aged 0–60 months and one out of every 28 deaths in children aged 28 days to 6 months [5]. Similarly, RSV can cause morbidity and mortality in adults. A systematic review of 103 studies published between 2000 and 2019 in developed countries evaluated the disease burden and healthcare utilization associated with RSV in older adults (aged ≥ 60 years) and high-risk individuals with comorbidities. The study found that RSV accounted for 4.7–7.8% of symptomatic respiratory infections in older adults, with a case fatality proportion of 8.2%. Among high-risk adults, RSV caused about 7.0–7.7% of symptomatic respiratory infections, with a case fatality proportion of 9.9% [16].

The healthcare impact of RSV infection rivals that of influenza in older adults, with a systematic review demonstrating comparable rates of hospitalization and mortality [17]. For instance, among hospitalized patients in the United States, RSV and influenza A resulted in comparable hospital stays, intensive care unit admission (15% and 12%, respectively), and mortality rates (8% and 7%, respectively) [18]. Notably, RSV infections were responsible for 11.4% of chronic obstructive pulmonary disease hospitalizations, 10.6% of pneumonia hospitalizations, 7.2% of asthma hospitalizations, and 5.4% of congestive heart failure hospitalizations [18]. More recent studies even suggest that RSV causes more severe disease than influenza, with greater rates of mechanical ventilation, intensive care unit admission, and mortality [19,20].

Further data highlight the substantial economic burden of RSV hospitalizations among older adults. According to a study conducted in Ontario, Canada, there were 7091 adults hospitalized with RSV between 2010 and 2019 [21]. The study found that the difference in healthcare costs between RSV-admitted patients and matched controls grew from $28,260 in the first 6 months to $43,721 at 2 years post-hospitalization.

Although RSV is important to individual and population health, even healthcare workers may have poor knowledge about RSV and RSV vaccination [22]. This narrative review, therefore, aims to serve as a practical primer for clinicians who are unfamiliar with RSV vaccination and will focus on two broad questions:What is the clinical course, investigation, and treatment of RSV infection?What are the relevant clinical studies and guideline recommendations supporting RSV vaccination?

With an improved understanding of both RSV infection and RSV vaccination, clinicians can then better guide patients regarding RSV vaccination.

## 2. Methods

PubMed^®^ (pubmed.ncbi.nlm.nih.gov) was used to search the MEDLINE database for contemporary literature over the past 3 years (from 1 October 2020 to 21 October 2023) and to update the author’s personal library of articles. The search terms were “(respiratory syncytial virus) AND (vaccination OR vaccine*)”. Titles and abstracts were screened to look for relevant articles that provided new insights for the narrative review.

## 3. Results

### 3.1. Results of the Literature Search

The search yielded 1164 articles, and 72 articles were included. Another 23 articles were used from the author’s personal library of articles. As such, 95 articles were included in this narrative review.

### 3.2. Clinical Course of RSV Infection

RSV commonly attacks the bronchopulmonary epithelium, causing lower respiratory tract disease, airway inflammation, sputum production, and bronchospasm. Symptoms include fever, cough, breathlessness, and wheezing [23]. These symptoms are non-specific and can occur with other viral respiratory infections. Symptoms peak in severity 5–7 days after illness onset. Within the first two weeks of infection, RSV can be recovered from the nasopharynx [24].

Extrapulmonary RSV infection can occur in both immunocompetent and immunocompromised patients, e.g., otitis media [25], encephalitis [26], and myocarditis [27]. In children aged < 15 years old, RSV is responsible for up to 6.5% of acute encephalitis, with 85% of cases presenting with seizure [26]. Other RSV-associated complications have also been reported in severely ill children who require mechanical ventilation, with elevated cardiac troponin levels in up to 54%, raised hepatic transaminases in up to 49%, and hyponatremia in about 33% [28].

Patients at risk for severe RSV lower respiratory tract disease are those with impaired immunity or reduced cardiorespiratory reserve. These patients include immunocompromised patients (e.g., solid organ transplant and hematopoietic stem cell transplant patients) [29,30], patients 65 years or older [31]), and patients with the following list of chronic conditions [32,33,34]:Active or past tobacco smokingDiabetes mellitusLung disease, e.g., asthma, chronic obstructive pulmonary diseaseHeart disease, e.g., congestive heart failureLiver diseaseRenal disease

Functional recovery after RSV infection may be incomplete. As an example, 39% of RSV-infected individuals residing in homeless shelters indicated that their ability to carry out their usual activities was significantly affected by their illnesses [35]. Separately, the protective humoral antibody response after natural RSV infection is short-lived; reinfection is possible in most people two months or more after natural infection [36].

### 3.3. Investigation of RSV Infection

As observed in chest computed tomography scans, the radiographic manifestations of RSV infection can encompass a range of features, such as localized or widespread interstitial or ground-glass pulmonary opacities, thickening of bronchial walls, tree-in-bud patterns, and lobar consolidation [37]. These findings lack specificity and may resemble those seen in influenza and parainfluenza viral infections, although it is worth noting that RSV infection tends to localize in the upper and middle lobes of the lung [37].

Given nonspecific clinical and radiographic findings of RSV infection, definitive diagnosis requires demonstration of live virus in culture, viral components (RSV antigen), positive acute viral serology, or viral RNA. Although viral culture is considered the gold standard, it is not feasible for clinical settings due to its extended turnaround time of up to one week and its limited sensitivity. Molecular methods are preferred, given high sensitivity, high specificity, and rapid turnaround times within hours [38]. Such molecular assays may even be available as point-of-care tests, which provide both convenience and timeliness for clinicians and patients [39]. Respiratory sampling for RSV is most easily carried out using nasopharyngeal or mid-turbinate nasal swabs. RSV samples can also be obtained from oropharyngeal swabs or from bronchoalveolar lavage.

According to a systematic review and meta-analysis of studies conducted between January 2000 and December 2021, singleplex reverse transcription polymerase chain reaction (RT-PCR) is the most sensitive diagnostic method for RSV in adults [40]. Using singleplex PCR as the reference standard (i.e., assumed sensitivity of 100%), other methods, such as rapid antigen detection test, direct fluorescent antibody, viral culture, and multiplex PCR, were less sensitive (pooled sensitivity 64%, 83%, 86%, 93%, respectively). The addition of various specimen types to nasopharyngeal swab RT-PCR testing significantly increased RSV detection rates, e.g., sputum RT-PCR increases detection by 1.5 times, and oropharyngeal RT-PCR increases detection by 1.3 times. Testing multiple specimens is, therefore, a useful strategy when the clinical suspicion for RSV infection is high but the initial nasopharyngeal PCR test is negative.

### 3.4. Treatment of RSV Infection

Supportive therapy is the primary approach for managing lower respiratory tract infection caused by RSV as specific antiviral therapy is not generally available. Ribavirin, a nucleoside analog, demonstrates good activity against RSV in laboratory settings [41]. Oral ribavirin has, however, not shown clinical benefits, while the aerosolized form (2 g over 3 h every 8 h, for 5–10 days [42]) has been considered for off-label use to prevent the progression of RSV upper respiratory tract infection to lower respiratory tract disease in immunocompromised individuals, e.g., hematopoietic stem cell transplant recipients [29,30].

EDP-938, an orally administered, small-molecule, non-fusion replication inhibitor targeting RSV nucleoprotein, holds significant potential as a treatment option. In a double-blind, randomized, placebo-controlled human virus challenge trial, it demonstrated superiority over a placebo in terms of reducing viral load, total symptom scores, and mucus weight while showing no apparent safety issues [43].

A novel nebulized trivalent nanobody (ALX-0171) with antiviral properties against RSV was studied in a double-blind, randomized, placebo-controlled, phase 2b trial in 50 hospital pediatric departments across 16 countries. However, despite decreasing viral load, ALX-0171 failed to improve oxygenation or disease severity in hospitalized children with RSV-related lower respiratory tract infection [44].

### 3.5. General Principles for Prevention of RSV Infection

Given the dearth of specific treatment options, especially after lower respiratory tract disease develops, prevention rather than treatment remains key to decreasing the health burden of RSV. As RSV is primarily transmitted by nasopharyngeal spread and ocular/mucous membrane inoculation, prevention of infection can be carried out by masking, cough hygiene, hand washing, and avoiding close contact with infected persons. Nosocomial transmission of RSV should be prevented via strict infection control, as outcomes of healthcare-associated RSV infection may be worse than in community-acquired infections. For instance, in a population-based, surveillance study of RSV-infected hospitalized adults, healthcare-associated infections required a higher level of care at discharge (i.e., placement in a skilled nursing or rehabilitation facility) compared with community-acquired infections (44% vs. 14%) [45].

Direct infant immunization is currently not available. Concerns exist over vaccine-enhanced RSV disease in patients without prior RSV infection [46], and proposed mechanisms encompass several pathways [47]. Firstly, antibody-dependent enhancement (ADE) involves vaccine-generated antibodies that fail to neutralize the virus. Secondly, immune complex accumulation and deposition in the lungs trigger an inflammatory environment, antibody production, and complement cascade activation, potentially worsening the disease. Thirdly, skewing the immune response toward a Th2 phenotype results in pulmonary eosinophil infiltration and disease exacerbation. Nonetheless, recent clinical trials in RSV-seronegative infants and children seem to suggest that vaccine-enhanced RSV disease may not always occur [48,49].

Until pediatric RSV vaccination is available, prevention of RSV disease in infants and children requires either immunoprophylaxis with monoclonal antibodies or transplacental transfer of maternal RSV antibodies following maternal vaccination during pregnancy. Palivizumab (Synagis^TM^, AstraZeneca/MedImmune) is a humanized monoclonal antibody that targets the RSV F glycoprotein [50]. Palivizumab is given via monthly intramuscular injections during the RSV season to protect children from severe lower respiratory tract infections caused by RSV [51]. Nirsevimab (Beyfortus^TM^, AstraZeneca/Sanofi) is another monoclonal antibody for RSV immunoprophylaxis that targets the prefusion conformation of the RSV F glycoprotein [52]. Nirsevimab possesses a prolonged half-life, and a single injection appears to be effective. Compared to the multiple monthly injections required for palivizumab, the single-dose regimen for nirsevimab makes it a more convenient monoclonal antibody option. Immunoprophylaxis with monoclonal antibodies is cost-effective in both high-income [53] and resource-limited countries [54,55].

### 3.6. Vaccination for Prevention of RSV Infection

To prevent RSV-associated lower respiratory tract infections in infants born during their first RSV season or entering it, administering maternal RSVpreF vaccination between 24 and 36 weeks of gestation during pregnancy is an alternative to providing immunoprophylaxis. Only one vaccine (Abrysvo^TM^, Pfizer Inc., New York, NY, USA) is available for this purpose [56], which contains stabilized preF from the two main RSV antigenic subgroups A and B. The RSV vaccine stimulates maternal production of protective antibodies against RSV, which takes about two weeks. Transplacental transfer of these antibodies then protects newborns and infants from severe RSV infection [57].

In the pivotal Phase 3 trial by Kampmann et al. [57], which recruited pregnant women between 24 and 36 weeks of gestation, numerically more cases of preterm births occurred in vaccinated compared to unvaccinated pregnant women (229 versus 198). Although the difference in the proportion of preterm births (6.4% versus 5.5%) was not statistically significant, to mitigate the risk of preterm births, the U.S. Food and Drug Authority approved maternal vaccination only for pregnant women between 32 and 36 weeks of gestation (Table 1). Similarly, both the American College of Obstetricians and Gynecologists and the U.S. Centers for Disease Control and Prevention limit their recommendations for maternal vaccination to 32–36 weeks of gestation (Table 2). Nonetheless, in contrast to the United States-based regulators, the European Medical Agency (EMA) licensed Abrysvo^TM^ on 23 August 2023 for maternal vaccination women in weeks 24–36 of gestation [58].

After maternal RSVpreF vaccination, immunoprophylaxis for infants is generally not needed [59]. However, special infant-related and maternal circumstances may necessitate immunoprophylaxis despite maternal RSVpreF vaccination [10]. Firstly, infants who are born within the first 14 days after maternal RSVpreF vaccination since a minimum of 14 days is required after maternal vaccination for the development. Secondly, infants who lose maternal antibodies due to cardiopulmonary bypass or extracorporeal membrane oxygenation. Thirdly, infants with especially high risk for severe RSV disease and who require enhanced protection from RSV, such as those with hemodynamically significant congenital heart disease or those with ongoing respiratory failure at hospital discharge. Fourthly, mothers with immunocompromise and who do not mount an adequate immune response to vaccination. Fifthly, mothers living with human immunodeficiency virus infection who have reduced transplacental antibody transfer.

Pregnant individuals can receive the maternal RSVpreF vaccine concurrently with other recommended vaccines, including tetanus, diphtheria, and pertussis (Tdap), influenza, and COVID-19 vaccines. The timing of administration, including the option for simultaneous vaccination at different anatomic sites on the same day, is flexible and not restricted [10]. Although RSVpreF and Tdap vaccines can be coadministered, the immunogenicity of the pertussis component may be blunted, though it seems to remain protective [60].

In contrast to infants, direct immunization of older adults is available, given that most adults would have been infected by RSV previously in their childhood or early adulthood. Two vaccines are currently available for adults aged 60 years and above: a non-adjuvanted bivalent vaccine (Abrysvo^TM^, Pfizer Inc., which is the same one used for maternal vaccination) and an adjuvanted non-bivalent vaccine (Arexvy^TM^, GSK Inc., London, UK) (Table 1 and Table 2).

**Table 1 vaccines-11-01809-t001:** Available types of RSV vaccines.

Vaccine	Target Group	Details of Vaccine Administration	Regulatory Approval by the U.S. FDA	Regulatory Approval by the EMA
RSVpreF vaccine (Abrysvo^TM^, Pfizer Inc.), a non-adjuvanted, bivalent recombinant stabilized prefusion F protein subunit vaccine	Infants (via maternal immunization at 32–36 weeks gestation for U.S. FDA, and at 24–36 weeks gestation for EMA)	Single intramuscular dose of 120 mcg (0.5 mL). Coadministration with Tdap, influenza, and COVID-19 vaccines is possible	21 August 2023 [10,59]	23 August 2023 [58]
RSVpreF vaccine (Abrysvo^TM^, Pfizer Inc.), a non-adjuvanted, bivalent recombinant stabilized prefusion F protein subunit vaccine	Individuals aged ≥ 60 years	Single intramuscular dose of 120 mcg (0.5 mL). Coadministration with influenza vaccine is possible	31 May 2023 [11]	23 August 2023 [58]
RSVpreF3 vaccine (Arexvy^TM^, GSK Inc.), an AS01E-adjuvanted, non-bivalent recombinant prefusion F protein subunit vaccine	Individuals aged ≥ 60 years	Single intramuscular dose of 120 mcg (0.5 mL). Coadministration with influenza vaccine is possible	3 May 2023 [11]	6 June 2023 [61]

COVID-19: Coronavirus disease 2019; EMA: European Medicines Agency; FDA: Food and Drug Administration; GSK: GlaxoSmithKline; RSV: Respiratory syncytial virus; Tdap: Tetanus, diphtheria, and pertussis; U.S.: United States.

**Table 2 vaccines-11-01809-t002:** Selected guideline recommendations for RSV vaccination.

Guideline Agency	Target Group	Details	Reference
American College of Obstetricians and Gynecologists (ACOG)	Pregnant women at 32 through 36 weeks gestation	Seasonal administration of one dose of RSV vaccine (Abrysvo, Pfizer Inc.) to prevent RSV LRTD in infants	ACOG (2023) [59]
U.S. Centers for Disease Control and Prevention	Pregnant women at 32 through 36 weeks gestation	Seasonal administration of one dose of RSV vaccine (Abrysvo, Pfizer Inc.) to prevent RSV LRTD in infants	Fleming-Dutra (2023) [10]
U.S. Centers for Disease Control and Prevention	Adults 60 years and older	Single dose of RSV vaccine with either Abrysvo^TM^, (Pfizer Inc.) or Arexvy^TM^, (GSK Inc.) to prevent RSV LRTD	Melgar (2023) [11]

GSK: GlaxoSmithKline; LRTD: Lower respiratory tract disease; RSV: Respiratory syncytial virus; U.S.: United States.

### 3.7. Evidence Supporting Efficacy and Safety of RSV Vaccination

This section provides the evidence that supports maternal RSV vaccination (for prevention of infection in infants) and RSV vaccination in older adults. The usual endpoint for clinical trials is the relative risk (RR) of developing symptomatic disease. Using data from a randomized clinical trial, the RR is calculated as the ratio of the probability of symptomatic disease in the vaccinated group to the probability of symptomatic disease in the unvaccinated group. Vaccine efficacy (VE) is then expressed as a percentage by subtracting the relative risk (RR) from 1 and then multiplying by 100, i.e., VE = (1 − RR) × 100%.

During the development of maternal RSV vaccination, RSVpreF vaccine candidates were first tested in healthy nonpregnant women to demonstrate immunogenicity (i.e., generation of serum-neutralizing responses) and safety (i.e., absence of major side effects or severe reactogenicity) [60,62,63] (Table 3). Then, trials of RSV vaccination in pregnant women [57,64,65] were conducted (Table 4). While developing the RSVpreF (non-adjuvanted, bivalent) vaccine (Abrysvo^TM^, Pfizer Inc.), studies found that pregnant women between 24 and 36 weeks of gestation appropriately developed neutralizing antibodies against both RSV-A and RSV-B following bivalent RSVpreF vaccination, but the humoral responses were not significantly different between lower (120 mcg) and higher (240 mcg) vaccine doses [64]. Furthermore, using an aluminum hydroxide adjuvant increased postvaccination reactions (e.g., infections, gastrointestinal conditions) without improving immunogenicity. As such, the final formulation of bivalent RSVpreF vaccine for maternal vaccination uses the 120 mcg dose without adjuvant [57].

Like in maternal RSV vaccination, during the development of RSV vaccination for older adults, RSVpreF vaccine candidates were shown to be immunogenic and safe in healthy young and older adults [56,66] (Table 3). The same formulation of the non-adjuvanted, bivalent RSVpreF vaccine (Abrysvo^TM^, Pfizer Inc.) is used in pregnant women and older adults aged 60 years and above [32] (Table 4). For the non-bivalent RSVpreF3 vaccine (Arexvy^TM^, GSK Inc.), the 120 mcg-dose with AS01E adjuvant had optimal immunogenicity and reactogenicity [66] and was thus used in the subsequent landmark clinical trial [33].

RSVpreF vaccination in pregnant [57] and older adults [32,33] is safe, with only mild-to-moderate side effects, such as injection site pain, myalgia, headache, and nausea. Although these vaccine-related reactogenicity events occur more frequently than when placebo vaccination is given, they resolve quickly within 2 to 3 days of onset and are, therefore, of minor clinical significance [57]. Nonetheless, continued vigilance for adverse events is required even for the approved vaccines. From the trial by Kampmann et al. [57], while RSVpreF vaccination had no direct adverse effects on infants after maternal vaccination, numerically more cases of preterm births occurred in vaccinated compared to unvaccinated pregnant women. This difference was not statistically significant. In addition, from the U.S. Centers for Disease Control and Prevention review of RSVpreF vaccination trials in older adults [11], rare cases of severe inflammatory neurological events (e.g., acute disseminated encephalomyelitis and Guillain–Barré syndrome) surfaced. Though concerning, these cases could not be definitively linked to RSV vaccination.

For pregnant women, coadministration of RSVpreF and a variety of other vaccines (seasonal inactivated influenza vaccine [67,68], Tdap vaccine [60]) is possible without major safety concerns. Although the immunogenicity of the pertussis component of Tdap may be blunted (the reason for this is unknown), it remains protective [60]. For older adults, coadministration of RSVpreF and influenza vaccine is possible, with good immunogenicity preserved for both vaccines, though reactogenicity may be increased [11].

**Table 3 vaccines-11-01809-t003:** Selected randomized clinical trials of RSV vaccines in healthy non-pregnant adults only (≥18 years old).

Target Group	Vaccine Type. Study Site (Country)	Trial Arms	Efficacy Outcome	Safety Outcome	Author (Year) [Reference]
Healthy men and nonpregnant women 18–49 years old	RSVpreF vaccine candidate (Pfizer Inc.). Study site: United States	565 received 60, 120, or 240 mcg RSVpreF vaccine with or without aluminum hydroxide adjuvant, 53 received placebo	All RSVpreF formulations generated protective virus-neutralizing titers one month after vaccination. Geometric mean fold rises across different RSVpreF doses and formulations ranged from 10.6 to 16.9 for RSV-A and from 10.3 to 19.8 for RSV-B	Recipients of RSVpreF reported local reactions and systemic events (mild to moderate) more often than those who received the placebo. No serious vaccine-related adverse events within 12 months following vaccination	Walsh (2022) [62]
Healthy nonpregnant women 18–49 years old	RSVpreF vaccine candidate (Pfizer Inc.). Study site: United States	286 received 120 or 240 mcg RSVpreF alone, 286 received 120 or 240 mcg RSVpreF with Tdap, 141 received placebo with Tdap	Combination of RSVpreF with Tdap showed no significant difference in the immune response against RSV-A and RSV-B compared to RSVpreF alone. Similarly, the immune response to diphtheria and tetanus toxoids when RSVpreF was given with Tdap was not inferior to that of Tdap alone. However, the noninferiority criterion was not met for the response to the pertussis component	Majority of reported reactogenicity events within one month of vaccination were of mild to moderate severity	Peterson (2022) [60]
Healthy nonpregnant women aged 18–45 years old	RSVpreF3 vaccine candidate (GSK Inc.). Study sites: United States, Finland, Germany	376 received a single intramuscular injection of unadjuvanted RSVPreF3 vaccine candidates (RSVPreF3 antigen concentrations 30–120 mcg), 126 received placebo	Following vaccination, the RSVPreF3 groups displayed a substantial increase in anti-RSV-A-neutralizing antibodies and anti-RSVPreF3 immunoglobulin G concentrations, with geometric mean titers and concentrations rising 8- to 14-fold and 12- to 21-fold, respectively, by day 8. These elevated levels remained significantly higher (5- to 6-fold and 6- to 8-fold) until day 91. Higher vaccine dose levels were more immunogenic than lower doses	Vaccinees experienced solicited local adverse events more frequently (ranging from 4% to 53.2%) compared to the placebo group (ranging from 0% to 15.9%). These events were of mild to moderate severity	Schwarz (2022) [63]
Healthy nonpregnant women 18–45 years old	Virus-like particle-based vaccine candidate (V306, Virometix). Study site: Belgium	45 received two intramuscular V306 at various doses, 15 received placebo	All vaccine doses were immunogenic with high RSV-neutralizing antibody titers	V-306 was found to be safe and well-tolerated across all dose levels, and there was no observed escalation in reactogenicity or unsolicited adverse events between the initial and subsequent administrations	Leroux-Roels (2023) [69]
Healthy adults 18–50 years old	RSVpreF vaccine candidate (Pfizer Inc.). Study site: United Kingdom	35 adults received 120 mcg RSVpreF vaccine, 35 adults received placebo	Vaccine efficacy of 86.7% for symptomatic RSV infection after intranasal virus challenge. About 28 days post-injection and just before the challenge phase, within the vaccinated group, the geometric mean factor increase in neutralizing geometric mean titers from the baseline was 20.5 for RSV-A and 20.3 for RSV-B	More local reactions (especially pain) were noted in the vaccine group than in the placebo group (14% vs. 6%). No serious adverse events were observed in either group	Schmoele-Thoma (2022) [56]
Healthy adults aged 18–40 years, and adults aged 60–80 years	RSVpreF3 vaccine candidate (GSK Inc.). Study sites: United States, Belgium	940 received one of nine RSVPreF3 formulations two doses two months apart (RSVPreF3 antigen concentrations 30–120 mcg, unadjuvanted or adjuvanted using AS01E or AS01B), 113 received placebo	Highest RSVPreF3 antigen level (120 mcg) with adjuvant had best immunological benefit	RSVPreF3 vaccine with AS01E adjuvant is less reactogenic than with AS01B adjuvant (AS01B has double the immunostimulant dose compared to AS01E). Although adjuvanted formulations were more reactogenic, adverse reactions were usually transient and not severe	Leroux-Roels (2023) [66]
Healthy younger adults (18–49 years) and healthy older adults (60–79 years)	mRNA-based vaccine candidate (Merck Inc., NJ, United States/Moderna Inc., MA, United States). Study site: Australia	155 received 25, 100, 200, or 300 mcg mRNA-V171 vaccine, 45 received placebo	Vaccination led to adequate humoral immune response (increases in RSV-neutralizing antibody titers, serum antibody titers to RSV prefusion F protein, D25 competing antibody titers to RSV prefusion F protein) and cell-mediated immune response (to RSV-F peptides)	All administered doses of vaccine were well-tolerated. No serious vaccine-related adverse events reported	Aliprantis (2021) [70]
Healthy younger adults (18–49 years) and healthy older adults (60–79 years)	mRNA-based vaccine candidate (Merck Inc./Moderna Inc.). Study site: United States	120 received 25, 100, 200, or 300 mcg mRNA-V172 vaccine, 40 received placebo	Vaccination led to adequate humoral immune response, i.e., elevated levels of serum-neutralizing antibodies, competing antibodies specific to pre-fusion F protein, and T-cell responses targeting RSV F-specific antigens	All administered doses of vaccine were well-tolerated. No serious vaccine-related adverse events reported	Nussbaum (2023) [71]
Healthy adults aged 18–45 years	G protein-based recombinant RSV (rRSV) vaccine candidate (BARS13, Advaccine Biopharmaceuticals Suzhou Co. Ltd., China). Study site: Australia	48 received various intramuscular doses of rRSV vaccine, 12 received placebo	In the high-dose group, serum-specific antibody levels were notably elevated, with a geometric mean concentration (GMC) of 885.74 IU/mL 30 days after the first dose, increasing to 1482.12 IU/mL 30 days after the second dose. These levels surpassed the GMC observed in the low-dose group, which had GMC values of 885.74 IU/mL after the first dose and 1187.10 IU/mL after the second dose	Within 30 days following each vaccination, none of the treatment-emergent adverse events resulted in participants discontinuing the study, and there were no reports of serious adverse events. Most recorded treatment-emergent adverse events were mild	Cheng (2023) [72]
Healthy adults aged 18–50 years	Recombinant *Mycobacterium bovis* BCG vaccine expressing the nucleoprotein of RSV (rBCG-N-hRSV) (Millennium Institute, Chile). Study site: Chile	18 received rBCG-N-hRSV vaccine, 2 received standard BCG vaccine	Notable rise in serum IgG-neutralizing antibodies targeting both Mycobacterium and the N-protein of RSV. All RSV vaccine recipients demonstrated an enhanced cellular response, characterized by the production of interfern-γ and IL-2, against PPD and the N-protein	Vaccine candidate was well-tolerated, without serious adverse events	Abarca (2020) [73]
Healthy adults aged 18–50 years	Ad26.RSV.preF vaccine candidate (J&J). Study site: United Kingdom	27 received a single intramuscular dose of Ad26.RSV.preF vaccine candidate, 26 received placebo	Following an intranasal virus challenge 28 days post-immunization, viral load, RSV infections (40.7% vs. 65.4%), and disease severity were lower in vaccinees compared to placebo recipients	Vaccine candidate was well-tolerated	Sadoff (2022) [74]
Healthy adults aged 18–50 years	Ad26.RSV.preF vaccine candidate (J&J). Study sites: United States, Finland, United Kingdom	8 received 2 intramuscular doses of Ad26.RSV.preF vaccine candidate at days 1 and 29, and 4 received placebo	Rise in RSV-A2-neutralizing antibodies, as well as in pre-F and post-F protein binding antibody titers, 28 days following the initial administration of Ad26.RSV.preF. After the second dose, these titers remained consistent for 28 days and exhibited a slight decline after 6 months	Among adults, the reported solicited adverse events were typically of mild to moderate intensity, and there were no instances of serious adverse events	Stuart (2022) [75]
Healthy adults aged ≥ 60 years old	Ad26.RSV.preF vaccine candidate (J&J). Study site: United States	49 received 1 or 2 intramuscular injections (12 months apart) of low or high dose Ad26.RSV.preF vaccine, 24 received placebo	After the initial vaccination, there were substantial increases in immune responses, including RSV-A2-neutralization titers, preF-specific antibodies, and the frequency of T cells secreting interferon-γ specific to F. These heightened immune responses were maintained at levels above the baseline for up to two years after immunization. Additionally, it was observed that a second vaccination at one year could further boost these immune responses	44% of individuals who received the vaccine reported experiencing expected, non-serious side effects. These side effects were temporary and of mild to moderate severity, and no severe adverse events were associated with the vaccination	Williams (2020) [76]
Healthy adults aged ≥ 55 years old	Multivalent live recombinant RSV vaccine candidate based on a nonreplicating Modified Vaccinia Ankara poxvirus (MVA-BN-RSV, Bavarian Nordic Inc., Denmark). Study site: Germany	337 received two intramuscular injections 4 weeks apart of MVA-BN-RSV at one of five doses, 83 received placebo	A single dose of the vaccine significantly increased both neutralizing and total antibodies while also stimulating a robust Th1-oriented cellular immune response against all five vaccine components. These elevated antibody responses were sustained for six months. A 12-month booster dose further enhanced antibody and T-cell responses	No serious vaccine-related adverse events were reported. NB. Development of MVAB-N-RSV vaccine halted in 2023 due to failure of a Phase 3 trial in 2023 (NCT05238025) [77]	Jordan (2021) [78]

NB. Table entries grouped by vaccine type and then sorted by date of publication. Ad26.RSV.preF: Replication-incompetent adenovirus 26 vector encoding the F protein stabilized in prefusion conformation; BCG: Bacillus Calmette–Guérin; BN: Bavarian Nordic; GSK: GlaxoSmithKline; J&J: Johnson & Johnson; LRTD: Lower respiratory tract disease; mRNA: Messenger ribonucleic acid; RCT: Randomized controlled trial. RSV: Respiratory syncytial virus; RSVpreF: RSV prefusion F protein; Tdap: Tetanus, diphtheria, and pertussis.

**Table 4 vaccines-11-01809-t004:** Selected randomized clinical trials of RSV vaccines not restricted to healthy nonpregnant adults (≥18 years old).

Target Group	Vaccine Type. Study Site (Country)	Trial Arms	Efficacy Outcome	Safety Outcome	Author (Year) [Reference]
Pregnant women at 24–36 weeks of gestation	RSVpreF vaccine candidate (Pfizer Inc.). Study sites: United States, Chile, Argentina, South Africa	327 women received RSVpreF vaccine (120 or 140 mcg, with or without aluminum hydroxide adjuvant), and 79 received placebo	Geometric mean ratios of 50% neutralizing titers between infants born to vaccine recipients and those born to placebo recipients ranged from 9.7 to 11.7 for individuals with RSV-A-neutralizing antibodies and from 13.6 to 16.8 for those with RSV-B-neutralizing antibodies. Transplacental transfer ratios of neutralizing antibodies varied from 1.41 to 2.10 and were higher when non-aluminum formulations were used as compared to aluminum formulations	Most postvaccination reactions (e.g., infections, gastrointestinal conditions) were mild to moderate and similar in the vaccine and placebo groups. The aluminum hydroxide adjuvant heightened the reactogenicity (local and systemic reactions) of the vaccine, but it did not provide any immunological benefits	Simoes (2022) [64]
Pregnant women at 24–36 weeks of gestation	RSVpreF vaccine (Abrysvo^TM^, Pfizer Inc.). Study sites: Argentina, Australia, Brazil, Canada, Chile, Denmark, Finland, Gambia, Japan, South Korea, Mexico, Netherlands, New Zealand, South Africa, Spain, Taiwan, China, United States	3682 received a single intramuscular injection of 120 mcg of RSVpreF vaccine without aluminum hydroxide adjuvant, 3676 received placebo	Medically attended severe lower respiratory tract illnesses occurred within 90 days after birth in fewer infants of women who received the vaccine (6 cases) compared to those in the placebo group (33 cases), resulting in a vaccine efficacy of 81.8%. Within 180 days after birth, the vaccine still showed effectiveness, with 19 cases in the vaccine group and 62 cases in the placebo group, resulting in a vaccine efficacy of 69.4%	Numerically, more cases of preterm births and hypertension/pre-eclampsia in vaccinated pregnant mothers but these were not statistically significant. Incidences of adverse events within one month after injection or after birth were similar between the vaccine and placebo groups, with 13.8% of women and 37.1% of infants in the vaccine group experiencing adverse events, compared to 13.1% and 34.5%, respectively, in the placebo group	Kampmann (2023) [57] (a.k.a. the MATISSE study)
Pregnant women aged 18–40 years old in late 2nd or 3rd trimesters	RSVpreF vaccine candidate (GSK Inc.). Study sites: Australia, Canada, Finland, France, New Zealand, Panama, South Africa, Spain, United States	145 received a single intramuscular dose (60 or 120 mcg), 68 received placebo	Vaccine generated strong immune responses, particularly in terms of RSV-A- and RSV-B-neutralizing antibodies and anti-RSVPreF3 IgG; 120 mcg dose appeared to be more immunogenic compared to the 60 mcg dose. Maternal antibodies were effectively passed on to infants and remained detectable for at least six months following birth	Vaccine demonstrated excellent tolerability, with no adverse events of particular interest related to pregnancy or neonatal outcomes considered attributable to the vaccine or placebo. NB. Development of vaccine has been halted due to increased preterm births noticed in further trials [79]	Bebia (2023) [65]
Pregnant women 28–36 weeks gestation	RSV fusion protein nanoparticle vaccine (Novavax). Study sites: Argentina, Australia, Chile, Bangladesh, Mexico, New Zealand, Philippines, South Africa, Spain, United Kingdom, United States	3051 received a single intramuscular dose of RSV fusion protein nanoparticle vaccine, 1585 received placebo	During the first 90 days of life, the vaccine group showed a lower percentage of infants with RSV-associated, medically significant lower respiratory tract infections at 1.5%, compared to the placebo group at 2.4% (vaccine efficacy of 39.4%, lower bound of 97.52% confidence interval of −1.0%, which did not meet prespecified success criterion)	Injection-site reactions at the local level were more frequent in women who received the vaccine compared to those who received the placebo (40.7% vs. 9.9%). However, the percentages of participants experiencing other adverse events were similar in both groups	Madhi (2020) (a.k.a. Prepare study) [80]
Adults 50–85 years old	RSVpreF vaccine candidate (Pfizer Inc.). Study site: United States	564 got 60, 120, or 240 mcg RSVpreF, with or without aluminum hydroxide adjuvant, with or without seasonal inactivated influenza vaccine (SIIV), 53 participants received placebo	All RSVpreF formulations, whether administered alone or with SIIV, resulted in strong RSV serum-neutralizing responses one month after vaccination. These neutralizing responses remained significantly elevated, 6.9–14.9 times at one month and 2.9–4.5 times at twelve months post-vaccination, compared to pre-vaccination levels. However, SIIV immune responses showed a tendency to be lower when administered alongside RSVpreF	In older adults, mild to moderate reactogenicity was observed in RSVpreF recipients, and adverse events within the first month post-vaccination were similar across different vaccine formulations. Coadministration with SIIV did not appear to affect safety in both younger and older adults	Falsey (2022) [67]
Adults aged ≥60 years	Ad26.RSV.preF vaccine candidate (J&J). Study site: Not stated	264 received various doses and combinations of Ad26.RSV.preF and recombinant RSVpreF protein intramuscular injections, 24 received placebo	Combination vaccination schedules elicited higher levels of humoral immune responses, including virus-neutralizing and preF-specific binding antibodies, as well as similar cellular immune responses (RSV-F-specific T cells) when compared to the use of Ad26.RSV.preF alone. These vaccine-induced immune responses were sustained above baseline levels for up to 1.5 years following vaccination	Each vaccine formulation was well-received, and all formulations exhibited comparable reactogenicity	Comeaux (2023) [81]
Adults aged ≥60 years	Ad26.RSV.preF vaccine candidate (J&J). Study site: United States	90 received Ad26.RSV.preF plus influenza vaccine on day 1 and placebo on day 29; 90 received placebo plus influenza vaccine on day 1 and Ad26.RSV.preF on day 29	Both trial groups exhibited strong neutralizing and binding antibody responses to RSV	Coadministration of Ad26.RSV.preF and influenza vaccine was well-tolerated	Sadoff (2021) [82]
Adults 65–85 years old	RSVpreF vaccine candidate (Pfizer Inc.). Study site: Australia	255 received 60, 120, or 240 mcg RSVpreF (with or without aluminum hydroxide adjuvant) alone or concomitantly with seasonal inactivated influenza vaccine (SIIV), 62 adults received placebo	All RSVpreF vaccine formulations produced strong and lasting serum-neutralizing responses. A second dose of RSVpreF vaccine 2 months after a 240 mcg dose did not enhance the response. RSVpreF and SIIV responses were similar or slightly lower when administered together	Mild adverse reactions were more common after RSVpreF compared to the placebo. Coadministration with SIIV did not appear to affect safety	Baber (2022) [68]
Adults aged 60–80 years in Japan	RSVpreF vaccine candidate (GSK Inc.). Study site: Japan	20 received AS01B-adjuvanted, non-bivalent vaccine (two 120 mcg doses two months apart), and 20 received placebo	Vaccine was immunogenic. RSVPreF3-specific IgG levels increased by 12.8 times on Day 31 and 9.2 times on Day 91. Neutralizing antibody titers increased by 7.3 times for RSV-A and 8.4 times for RSV-B on Day 31, and 6.3 times for RSV-A and 9.9 times for RSV-B on Day 91	Adverse events were reported more frequently in vaccinated persons (80–90%) than in placebo recipients (10–20%)	Kotb (2023) [83]
Adults aged 60 years and older	RSVpreF vaccine (Abrysvo^TM^, Pfizer Inc.). Study sites: Argentina, Canada, Finland, Japan, Netherlands, South Africa, United States	17,215 received a single intramuscular injection of 120 mcg of RSVpreF vaccine without aluminum hydroxide adjuvant, 17,069 received placebo	Vaccine efficacy of 66.7% for RSV-associated lower respiratory tract illness with at least two signs or symptoms and 85.7% for illness with at least three signs or symptoms	Vaccine group had a greater occurrence of local reactions (12%) in contrast to the placebo group (7%). Both groups exhibited similar rates of systemic events (27% in the vaccine group and 26% in the placebo group)	Walsh (2023) [32] (a.k.a. the RENOIR study)
Adults aged 60 years and older	RSVpreF3 vaccine (Arexvy^TM^, GSK Inc.). Study sites: Belgium, Canada, Estonia, Finland, Germany, Italy, Japan, Mexico, Poland, Russia, Spain, South Korea, United Kingdom, United States, Australia, New Zealand, South Africa	12,467 received a single intramuscular dose of vaccine (containing 120 mcg PreF3 antigen with 50 mcg AS01E adjuvant), and 12,499 received placebo	During a median follow-up period of 6.7 months, the vaccine showed an efficacy of 82.6% against all RSV-LRTD and 94.1% against severe RSV-LRTD. The vaccine performed consistently well against both RSV-A and RSV-B subtypes and among participants with coexisting medical conditions	Greater reactogenicity for vaccine compared to the placebo, although the adverse events were mostly transient and mild to moderate in severity	Papi (2023) [33] (a.k.a. the AReSVi-006 study)
Adults aged 65 years and older	Ad26.RSV.preF vaccine candidate (J&J). Study site: United States	2891 received a single Ad26.RSV.preF and recombinant RSVpreF protein intramuscular injection, 2891 received placebo	RSV-induced lower respiratory tract disease was defined in three ways: three or more lower respiratory symptoms (definition 1), two or more lower respiratory symptoms (definition 2), a combination of two or more lower respiratory symptoms, or one or more lower respiratory symptoms along with at least one systemic symptom (definition 3). Vaccine efficacy rates were 80.0%, 75.0%, and 69.8% for these three case definitions, respectively. RSV-A2-neutralizing antibody levels increased by a factor of 12.1 from the baseline to day 15 post-vaccination	Vaccine group had a higher percentage of participants experiencing adverse events compared to the placebo group (local, 37.9% vs. 8.4%; systemic, 41.4% vs. 16.4%). Most of these adverse events were of mild to moderate severity. NB. Development halted by manufacturer in March 2023 after portfolio review	Falsey (2023) [84] (a.k.a. the CYPRESS study)

NB. Table entries grouped by vaccine type and then sorted by date of publication. Ad26.RSV.preF: Replication-incompetent adenovirus 26 vector encoding the F protein stabilized in prefusion conformation; GSK: GlaxoSmithKline; J&J: Johnson & Johnson; LRTD: Lower respiratory tract disease; mRNA: Messenger ribonucleic acid; RCT: Randomized controlled trial. RSV: Respiratory syncytial virus; RSVpreF: RSV prefusion F protein; Tdap: Tetanus, diphtheria, and pertussis.

## 4. Discussion

### 4.1. Limitations of This Narrative Review

This review used primarily the MEDLINE database, which contains literature published predominantly in the English language and may, therefore, miss out studies published in other languages. In addition, the inclusion of studies is subjective, though the review is meant to summarize key information from the clinical perspective rather than to systematically review all literature. Finally, the efficacy and safety of RSV vaccination in special populations, such as immunocompromised patients, are extrapolated from available data in immunocompetent patients, and further studies are required.

### 4.2. Future Directions

The year 2023 was the breakthrough year for RSV vaccination, with the publication of three pivotal trials leading to the authorization of two vaccines for vaccination of pregnant women and adults aged ≥ 60 years [32,33,57], though more trials are needed to elucidate the protective effect of vaccination on severe RSV outcomes, such as respiratory failure, extrapulmonary complications, functional outcomes, quality of life, and mortality. RSV vaccination should be further studied in other patient groups at risk for severe RSV disease, e.g., younger adults and children with immunocompromise [29] and chronic cardiopulmonary disease [34]. For these younger adults, candidate vaccines have already demonstrated immunogenicity and safety in healthy individuals [56,66] and in the general population [67], while efficacy data are being accrued [56]. For children and infants, studies of vaccination in both RSV-seronegative and RSV-seropositive individuals are underway, with candidate vaccines showing immunogenicity and safety (Table 5). For immunocompromised persons of varying levels of immunocompromise, immunogenicity will be blunted to varying degrees [85], and dedicated vaccination studies will be needed to elucidate the appropriate vaccine dosing regimen (including dose and need for repeat vaccination). Given that RSV-associated acute respiratory infections affect about 3.4% of pregnant women, with increased odds of preterm delivery [86], it will also be important to know if adult vaccination of pregnant mothers can benefit pregnancy outcomes.

Multiple vaccine platforms are being developed for RSV vaccination and for various target groups [87]. Virus-like particle-based vaccines [69], mRNA-based vaccines [70,71], G protein-based recombinant vaccines [72], *Mycobacterium bovis* BCG-based recombinant vaccines [73], adenovirus vector-based vaccines [74,75,76] (development has been halted by its manufacturer in March 2023 after portfolio review), and Modified Vaccinia Ankara (MVA) poxvirus-based recombinant vaccines [78] have been immunogenic and well-tolerated in early trials in healthy participants. Beyond the trials in healthy participants, although a large multinational trial of RSV fusion protein nanoparticle vaccine did not reach its desired efficacy outcome [80], other platforms involving adenovirus vector-based vaccines [81,82,84] and mRNA-based vaccines appear more promising. The ConquerRSV trial, examining Moderna’s mRNA-based RSV vaccine (NCT05127434), enrolled over 37,000 participants 60 years of age and older and attained its primary efficacy endpoint of preventing symptomatic RSV-associated lower respiratory tract disease; results regarding the prevention of severe disease are anticipated [88].

**Table 5 vaccines-11-01809-t005:** Selected randomized clinical trials of RSV vaccines in children (<18 years old).

Target Group	Vaccine Type. Study Site (Country)	Trial Arms	Efficacy Outcome	Safety Outcome	Author (Year) [Reference]
RSV-seronegative children aged 6–24 months	Seven intranasal live attenuated RSV vaccine candidates (NIH). Study site: United States	160 received vaccine, 79 received placebo	Serum RSV-neutralizing antibodies following a single vaccine dose showed similarities to primary responses to wild-type RSV. The vaccine-induced RSV plaque reduction neutralization test (PRNT) results were long-lasting. Live-attenuated RSV vaccines that trigger at least a fourfold increase in RSV PRNT offered protection against medically attended acute respiratory illness and significant protection against medically attended acute lower respiratory illness	Not reported	Karron (2021) [89]
RSV-seronegative children aged 6–24 months	Two intranasal live attenuated RSV vaccine candidates (NIH). Study site: United States	52 received a single dose of intranasal live attenuated RSV vaccine, 13 received placebo	Serum RSV-neutralizing titers and anti-RSV F IgG titers showed a 4-fold or more increase in 60–92% of vaccine recipients	During the 28 days following inoculation, upper respiratory illness and/or fever were observed in 64–84% of vaccine recipients vs. 58% of placebo recipients. These symptoms were mild. No cases of lower respiratory illness or serious adverse events were reported	Cunningham (2022) [48]
RSV-seronegative children aged 6–24 months	Intranasal live attenuated (by deletion of interferon antagonist NS2 gene) RSV vaccine candidate (NIH). Study site: United States	20 received a single dose of intranasal c-DNA derived live attenuated RSV vaccine, 10 received placebo	Vaccine elicited an immune response and maintained genetic stability in children who had not previously been exposed to RSV	In RSV-seronegative children, rhinorrhea occurred more frequently in vaccinated versus placebo recipients	Karron (2023) [49]
Healthy infants aged 6–7 months	Chimpanzee-derived adenoviral vector RSV vaccine candidate (ChAd155-RSV, GSK Inc.). Study sites: Brazil, Canada, Colombia, Finland, Italy, Mexico, Panama, Poland, Spain, Thailand, Turkey, United Kingdom, United States	201 received two injections of low/high dose of vaccine on days 1 and 31; 65 received placebo/active comparator vaccine *; 159 were RSV-seronegative at baseline	Levels of RSV-A-neutralizing titers and RSV F-binding antibody concentrations were greater after the administration of ChAd155-RSV compared to after the administration of the comparator vaccine on days 31, 61, and the end of 1st RSV season (averaging a follow-up of 7 months). High-dose ChAd155-RSV produced more robust responses than the low-dose, with additional increases observed after the second dose	Frequency of most anticipated and spontaneous adverse events following ChAd155-RSV vaccination was comparable to or even lower than those observed following the administration of active comparator vaccines. Among infants who contracted RSV infection, there were no indications of vaccine-enhanced disease	Saez-Llorens (2023) [90]
RSV-seropositive children aged 12–23 months old	Chimpanzee-derived adenoviral vector RSV vaccine candidate (ChAd155-RSV, GSK Inc.). Study sites: Canada, Italy, Mexico, Panama, Spain, Taiwan, United States	82 received two injections of various doses of vaccine at days 1 and 31; 39 received placebo	RSV-A-neutralizing antibody (Nab) titers showed dose-dependent increase after the first dose, with no additional rise after the second dose. RSV-A Nab titers remained elevated even one year after	Solicited adverse events were comparable between the groups, except for the occurrence of mild fever (<38.5 °C), which was more frequent in the highest dose group. No serious adverse events linked to the vaccine or RSV-related hospitalizations	Diez-Domingo (2023) [91]
RSV-seropositive children aged 12–24 months	Ad26.RSV.preF vaccine candidate (J&J). Study sites: United States, Finland, United Kingdom	24 received 2 intramuscular doses of Ad26.RSV.preF vaccine candidate at days 1 and 29; 12 received placebo	Significant increase in pediatric geometric mean titers for RSV-A2 neutralization from baseline to day 57 after receiving the Ad26.RSV.preF vaccine, with sustained elevated levels over 7 months. Additionally, fewer children who received the vaccine (4.2%) experienced confirmed RSV infection compared to those who received a placebo (41.7%)	No serious adverse events related to vaccination were reported, but 14 (58.3%) of the Ad26.RSV.preF recipients experienced fever	Stuart (2022) [75]

NB. Table entries grouped by vaccine type and then sorted by date of publication. * 1 of 4 comparator vaccines: Four-component meningococcal serogroup B vaccine (4CMenB, Bexsero, GSK) or meningococcal serogroups A, C, W, Y tetanus toxoid conjugate vaccine (MenACWY-TT, Nimenrix, Pfizer) or pneumococcal non-typeable *Haemophilus influenzae* protein D conjugate vaccine (PHiD-CV, Synflorix, GSK) or meningococcal serogroups A, C, W, Y CRM197 conjugate vaccine (MenACWY-CRM, Menveo, GSK). GSK: GlaxoSmithKline; J&J: Johnson & Johnson; NIH: National Institutes of Health; RSV: Respiratory syncytial virus.

For vaccination to be considered a success, vaccines need to reach their intended recipients. This objective can be achieved through the implementation of various strategies, which may encompass the following actions:Including RSV vaccination in global guidelines, especially those focusing on at-risk patients with chronic diseases [92].Including RSV vaccination in national policies and vaccination schedules, which would require not only efficacy/safety data, but also cost-effectiveness/economic impact data [93]. Special consideration should also be given to vaccination’s association with reduced disease and consequent antimicrobial prescribing [94].Improving access and affordability, especially in low-income and middle-income countries (LMICs), where over 95% of cases of RSV-associated acute lower respiratory infections and more than 97% of RSV-related deaths across all age groups occurred in children aged five years or younger [5].Addressing vaccine hesitancy, which is “the delay in acceptance or refusal of vaccines despite availability of vaccination services”, as defined by the World Health Organization’s Strategic Advisory Group of Experts on Immunization (SAGE).

Vaccine hesitancy is influenced by complacency, convenience, and confidence [95]. To overcome complacency (diminished perception of the risk associated with vaccine-preventable diseases), healthcare providers need to convince at-risk individuals about the severity of RSV disease. To overcome inconvenience, health systems need to make RSV vaccination affordable and easily available. Coadministration of RSV vaccines with influenza vaccine seems to be safe and may be a useful means to improve convenience. To overcome the lack of confidence, more clinical trial and surveillance data will be needed to demonstrate an improvement in key patient outcomes (e.g., survival and quality of life) and absence of excess risk for severe adverse events (e.g., preterm births [79], acute disseminated encephalomyelitis and Guillain–Barré syndrome [11]). Recent vaccine acceptance surveys for maternal RSV vaccination have highlighted the need for layperson education to build trust in healthcare providers, provide evidence on vaccine efficacy, and address safety concerns. In addition, vaccines should be provided at a reasonable cost to reduce financial burden [96]. In line with the aim of this narrative review, healthcare providers also need to be educated on RSV vaccination to improve the former’s awareness and support for routine vaccination [97,98].

## 5. Conclusions

This review underscores the significant impact of RSV on global health, particularly in infants, young children, and older adults. It highlights the importance of timely diagnosis, supportive care, potential antiviral treatments, and vaccination for reducing the health burden of RSV infection. Maternal vaccination for infant protection and vaccination in older adults are now available, with ongoing efforts to extend vaccination to younger at-risk populations and to seronegative children. For RSV vaccination to be implemented successfully, global guidelines and national policies need to recommend RSV vaccination, and health systems need to ensure the affordability and accessibility of RSV vaccines, especially in low- and middle-income countries. Finally, more trial data and post-marketing surveillance of RSV vaccines will help build confidence in the long-term efficacy and safety of RSV vaccination.

## Data Availability

All data used can be found in the text and tables.

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
