# Peer review of "Vaccination for Respiratory Syncytial Virus: A Narrative Review and Primer for Clinicians"

_vaccines, 2023, doi:10.3390/vaccines11121809_

Round 1
Reviewer 1 Report
Comments and Suggestions for Authors
The topic that was presented is important and timely for understanding the complexity of vaccination for Respiratory Syncytial Virus. However, there are significant changes that need to be made.
1) I find the structure you chose for this manuscript to be unappealing. In my opinion, the structure of narrative literature reviews could be similar to the traditional robust structure IMRDC, which encompasses Introduction, Methods, Results, Discussion, Conclusion. For instance, the section entitled 'Virology and epidemiology of RSV' could be part of 'Introduction', along with other sections like subchapters.
2) It is important to clarify the objective of the narrative review. Please explain the “practical primer for clinicians who are unfamiliar with RSV vaccination”.
3) The methodology and research strategy used are inadequate and require to be improved. When the keywords were used as indicated, the search yielded around four thousand results. What method did you use to identify the mentioned articles?
4) To improve the readability of Tables 3-4-6, it may be beneficial to use a horizontal orientation of the sheet. Furthermore, the tables need to contain the info about the country where the randomized clinical trial was carried out, as well as the sample size in a specific column that distinguishes the target group from the aim. Nonetheless, additional analysis of subgroups is necessary.
5) In section 9 “Future directions for RSV vaccination” the discussion about vaccine hesitation should be further developed if the author wants to address it, and it should be part of the narrative review citing the available literature.
Author Response
Reviewer 1
The topic that was presented is important and timely for understanding the complexity of vaccination for Respiratory Syncytial Virus. However, there are significant changes that need to be made.
1) I find the structure you chose for this manuscript to be unappealing. In my opinion, the structure of narrative literature reviews could be similar to the traditional robust structure IMRDC, which encompasses Introduction, Methods, Results, Discussion, Conclusion. For instance, the section entitled 'Virology and epidemiology of RSV' could be part of 'Introduction', along with other sections like subchapters.
Reply: The manuscript is restructured as suggested.
2) It is important to clarify the objective of the narrative review. Please explain the “practical primer for clinicians who are unfamiliar with RSV vaccination”.
Reply: The last paragraph of the introduction is re-written to specify the objective of the review. “Although RSV is important to individual and population health, even healthcare workers may have poor knowledge about RSV and RSV vaccination. This narrative review therefore aims to serve as a practical primer for clinicians who are unfamiliar with RSV vaccination and will focus on two broad areas: firstly, clinical course, investigation, and treatment of RSV infection; secondly, relevant clinical studies and guideline recommendations of RSV vaccination. With an improved understanding of both RSV infection and RSV vaccination, clinicians can then better guide patients with regards to RSV vaccination.”
3) The methodology and research strategy used are inadequate and require to be improved. When the keywords were used as indicated, the search yielded around four thousand results. What method did you use to identify the mentioned articles?
Reply: The search – updated on 21 Oct 2023 – for articles published from 1 Oct 2020 to 21 Oct 2023 yielded 1,164 articles. The URL is as follows: https://pubmed.ncbi.nlm.nih.gov/?term=%28respiratory+syncytial+virus%29+AND+%28vaccination+OR+vaccine*%29&filter=dates.2020%2F10%2F1-2023%2F10%2F21&sort=date&size=200. (NB. Using this same URL on 13 Nov 2023, I get a slightly different number of 1,169 articles instead of 1,164). Titles and abstracts were screened to look for relevant articles that provided new insights for the narrative review. This is admittedly subjective, and I include this as a limitation. I have also updated the methods and results to reflect the literature search and screening process.
4) To improve the readability of Tables 3-4-6, it may be beneficial to use a horizontal orientation of the sheet. Furthermore, the tables need to contain the info about the country where the randomized clinical trial was carried out, as well as the sample size in a specific column that distinguishes the target group from the aim. Nonetheless, additional analysis of subgroups is necessary.
Reply: Tables 3, 4, and 5 (re-numbered from 6) have been changed to the landscape orientation. Added a column for trial arms. Included study site (country). Table entries sorted by vaccine type and then by year of publication, with sorting order mentioned in the footnote. Important subgroups like immunocompromised patients are not well-studied in existing trials, and a note has been added to the limitations section (4.1): “Finally, efficacy and safety of RSV vaccination in special populations such as immuno-compromised patients are generally extrapolated from available data in immunocompetent patients, and further studies are required”.
5) In section 9 “Future directions for RSV vaccination” the discussion about vaccine hesitation should be further developed if the author wants to address it, and it should be part of the narrative review citing the available literature.
Reply: The discussion on RSV vaccination hesitancy has been improved with the addition of more recent references.
Reviewer 2 Report
Comments and Suggestions for Authors
Thank you for giving me the opportunity to review this manuscript. This narrative review provides useful guidance for clinicians on the important topic of new vaccines against Respiratory Syncytial Virus (RSV). With the first two RSV vaccines licensed in 2023, this is a timely contribution to the debate on RSV prevention in public health, particularly for clinicians in low- and middle-income countries.
The manuscript is well-written and clear. However, the overall article lengths should be reduced, especially regarding the extensive tables.
Detailed comments:
Abstract
The smoothness of this paragraph gives me the impression that it may have been written by AI. Please rephrase this section and list all AI tools used for the rest of the manuscript, if any.
1. Introduction
Please add a concrete research question.
2. Methods
This part does not seem very helpful to me. To my understanding, a narrative review does not require a methods section because the research question is addressed in a non-systematic way. In contrast, I would expect a more sophisticated method and additional information to be provided in a systematic review. For example, more than one medical database should be searched (e.g., PubMed and EMBASE), authors should use MeSH or other systematic terms, and the search window, language, and scope of the search should be justified. Additionally, I would expect a short quantitative description of the results, how many papers were excluded and included, how many were not available for fulltext review, the distribution of publication years, etc. In the case of this manuscript, I recommend removing the methods section entirely.
3. Virology and epidemiology of RSV
This section provides important information but should be shortened.
Line 54, Reference 13: Please cite a recent global surveillance or disease burden modeling study on RSV instead of another narrative review.
Line 62, Reference 15: Please cite the latest global under-5-mortality estimation study available.
Line 62: Please write out US.
Line 63: Please add a unit to “per 1000”.
Line 69: Please change “there were 33 million cases” to “there were an estimated 33 million cases”.
Line 78: Please use the abbreviation RSV.
Lines 85-87: Please add the source.
Lines 94-95: Please add the source.
4. Clinical course of RSV infection
Line 107, Reference 13: Please cite a clinical study or treatment guideline instead of another narrative review.
5. Investigation of RSV infection
Lines 127-130: Please add the source.
Line 145: Please give the number of the sensitivity of RT-PCR so that the reader can compare it to the less sensitive methods on the next page.
6. Treatment of RSV infection
Lines 154-159: Please directly cite clinical treatment guidelines here.
Line 160: Please add a citation on the activity of ribavirin against RSV in laboratory settings.
Line 172: Please add citations on the benefits and availability of RSV-IVIG and interferon-alpha.
Line 174: Please replace “unfortunately” by “however”.
7. General principles for prevention of RSV infection including vaccination
This section is very comprehensive. Please shorten it.
Line 206: Is this true for all RSV vaccines currently under development?
Line 207: Please change “Palivizumab a humanized monoclonal antibody” to “Palivizumab is a humanized monoclonal antibody”.
Line 212: Please add the source.
Line 230, Reference 51: This study had three first authors. Please change “Kampmann” to “Kampmann, Madhi, Munjal et al.”.
Line 235: The European Medical Agency (EMA) licensed Abrysvo (tm) on 25 August 2023 for women in weeks 24-36 of gestation, see https://www.ema.europa.eu/en/medicines/human/EPAR/abrysvo. Please add this information.
Line 240, Reference 51: Please cite a relevant public health recommendation or guideline rather than a clinical vaccine trial.
Line 257: Please change “remains protective” to “seems to remain protective”.
8. Evidence supporting efficacy and safety of RSV vaccination
Line 309, Reference 51: This study had three first authors. Please change “Kampmann” to “Kampmann, Madhi, Munjal et al.”.
Line 310: Please change “has” to “had”.
Line 312: Please remove “nonetheless”.
Line 315: Please change “Guillain-Barre” to “Guillain-Barré”.
Lines 320-321: Please add the source.
9. Future directions for RSV vaccination
Line 338: Please change “2023 was” to “The year 2023 was”.
Lines 341-343: Please add the sources for these groups.
Line 343: Please change “renal disease” to “renal disease patients”.
Line 344: The age limit of 18-64 years is unclear to me. Please justify or modify.
Line 367: Please change “modified vaccinia Ankara” to “Modified Vaccinia Ankara (MVA)”.
10. Conclusions
The smoothness of this paragraph gives me the impression that at least parts of it may have been written by AI. Please rephrase this section and list all AI tools used for the rest of the manuscript, if any.
Line 404: Please remove “In conclusion”.
Line 413: Please remove “)”.
Table 1
Column 1: Please add more information on the functional mechanism of the vaccines, e.g., bivalent recombinant protein subunit vaccine.
Please add a column on the regulatory approval by the European Medical Agency (EMA) and a column with references to the FDA and EMA approvals, respectively.
Please add the abbreviations RSV, COVID-19, Tdap, GSK, and U.S. to the subscripts of the table.
Table 2
Please add the abbreviations RSV, GSK, and U.S. to the subscripts of the table.
Table 3
This table is very bulky. However, it cannot entirely capture the complex information on each study (e.g, study design, primary and secondary outcomes, etc). Please consider extending the table by adding relevant columns or shortening it. Please sort by year of publication (descending).
Please change “modified vaccinia Ankara” to “Modified Vaccinia Ankara (MVA)”.
Please add the abbreviations GSK, NB, and J&J to the subscripts of the table.
Line 329: Please change “Bacillus Calmette-Guerin” to “Bacillus Calmette–Guérin”.
Table 4
This table is very bulky. However, it cannot entirely capture the complex information on each study (e.g, study design, primary and secondary outcomes, etc). Please consider extending the table by adding relevant columns or shortening it. Please sort by year of publication (descending).
Please add the abbreviations GSK, NB, and J&J to the subscripts of the table.
Table 5
This table adds very little additional information for the reader. I recommend removing it.
Table 6
This table is very bulky. However, it cannot entirely capture the complex information on each study (e.g, study design, primary and secondary outcomes, etc). Please consider extending the table by adding relevant columns or shortening it. Please sort by year of publication (descending).
Please add the abbreviations GSK, and J&J to the subscripts of the table.
Line 359: Please change “Haemophilus influenzae” to “haemophilus influenzae” (in italic).
Author Response
Reviewer 2
Thank you for giving me the opportunity to review this manuscript. This narrative review provides useful guidance for clinicians on the important topic of new vaccines against Respiratory Syncytial Virus (RSV). With the first two RSV vaccines licensed in 2023, this is a timely contribution to the debate on RSV prevention in public health, particularly for clinicians in low- and middle-income countries.
The manuscript is well-written and clear. However, the overall article lengths should be reduced, especially regarding the extensive tables.
Reply: The tables have been revised as recommended below.
Detailed comments:
Abstract
The smoothness of this paragraph gives me the impression that it may have been written by AI. Please rephrase this section and list all AI tools used for the rest of the manuscript, if any.
Reply: No part of the manuscript was written by AI. I wrote the entire draft and used the built-in Editor function of MS Word to improve the spelling, grammar, and language of the entire manuscript. Nonetheless, I re-read the abstract, and improved the smoothness by including a more active voice.
1. Introduction
Please add a concrete research question.
Reply: Two broad questions to guide the review have been added in the Introduction.
2. Methods
This part does not seem very helpful to me. To my understanding, a narrative review does not require a methods section because the research question is addressed in a non-systematic way. In contrast, I would expect a more sophisticated method and additional information to be provided in a systematic review. For example, more than one medical database should be searched (e.g., PubMed and EMBASE), authors should use MeSH or other systematic terms, and the search window, language, and scope of the search should be justified. Additionally, I would expect a short quantitative description of the results, how many papers were excluded and included, how many were not available for fulltext review, the distribution of publication years, etc. In the case of this manuscript, I recommend removing the methods section entirely.
Reply: I agree that narrative reviews are non-systematic, and I do not have the details that a full systematic review will require. Nonetheless, I had to retain and improve this section for my narrative review at the recommendation of another reviewer.
3. Virology and epidemiology of RSV
This section provides important information but should be shortened.
Reply: Less important information removed, and section shortened.
Line 54, Reference 13: Please cite a recent global surveillance or disease burden modeling study on RSV instead of another narrative review.
Reply: Global study by Chadha et al in Influenza Other Respir Viruses 2020 cited.
Line 62, Reference 15: Please cite the latest global under-5-mortality estimation study available.
Reply: Global estimate cited. United States and China references removed.
Line 62: Please write out US.
Reply: Global estimate cited. United States reference removed.
Line 63: Please add a unit to “per 1000”.
Reply: Global estimate cited. United States reference removed.
Line 69: Please change “there were 33 million cases” to “there were an estimated 33 million cases”.
Reply: Amendment done.
Line 78: Please use the abbreviation RSV.
Reply: Abbreviation used.
Lines 85-87: Please add the source.
Reply: Reference added.
Lines 94-95: Please add the source.
Reply: Reference added.
4. Clinical course of RSV infection
Line 107, Reference 13: Please cite a clinical study or treatment guideline instead of another narrative review.
Reply: Clinical study by Munywoki et al in Epidemiol Infect 2015 cited instead.
5. Investigation of RSV infection
Lines 127-130: Please add the source.
Reply: Reference added.
Line 145: Please give the number of the sensitivity of RT-PCR so that the reader can compare it to the less sensitive methods on the next page.
Reply: As the reference standard, sensitivity of RT-PCR is assumed to be 100%. This assumption is now stated in the text.
6. Treatment of RSV infection
Lines 154-159: Please directly cite clinical treatment guidelines here.
Reply: There are no clinical treatment guidelines that I can find for RSV. As such, I will remove this section.
Line 160: Please add a citation on the activity of ribavirin against RSV in laboratory settings.
Reply: Reference by Hruska et al published in Antimicrob Agents Chemother 1980 added.
Line 172: Please add citations on the benefits and availability of RSV-IVIG and interferon-alpha.
Reply: There are no citations available. As such, I will remove this statement.
Line 174: Please replace “unfortunately” by “however”.
Reply: Word replaced.
7. General principles for prevention of RSV infection including vaccination
This section is very comprehensive. Please shorten it.
Reply: Less important information has been removed.
Line 206: Is this true for all RSV vaccines currently under development?
Reply: This is only true for the licensed RSV vaccines. To avoid misleading readers, this statement is removed.
Line 207: Please change “Palivizumab a humanized monoclonal antibody” to “Palivizumab is a humanized monoclonal antibody”.
Reply: Change made.
Line 212: Please add the source.
Reply: The AAP 2023 guidance is added.
Line 230, Reference 51: This study had three first authors. Please change “Kampmann” to “Kampmann, Madhi, Munjal et al.”.
Reply: Change made.
Line 235: The European Medical Agency (EMA) licensed Abrysvo (tm) on 25 August 2023 for women in weeks 24-36 of gestation, see https://www.ema.europa.eu/en/medicines/human/EPAR/abrysvo. Please add this information.
Reply: Information added.
Line 240, Reference 51: Please cite a relevant public health recommendation or guideline rather than a clinical vaccine trial.
Reply: The ACOG 2023 guidance is added.
Line 257: Please change “remains protective” to “seems to remain protective”.
Reply: Change made.
8. Evidence supporting efficacy and safety of RSV vaccination
Line 309, Reference 51: This study had three first authors. Please change “Kampmann” to “Kampmann, Madhi, Munjal et al.”.
Reply: Change made.
Line 310: Please change “has” to “had”.
Reply: Change made.
Line 312: Please remove “nonetheless”.
Reply: Word removed.
Line 315: Please change “Guillain-Barre” to “Guillain-Barré”.
Reply: Change made.
Lines 320-321: Please add the source.
Reply: Lines 320-321 are meant to indicate the same information in the following two sentences about pertussis and influenza. They are therefore removed to shorten the text.
9. Future directions for RSV vaccination
Line 338: Please change “2023 was” to “The year 2023 was”.
Reply: Change made.
Lines 341-343: Please add the sources for these groups.
Reply: Data for RSV vaccination of these risk groups have not yet been developed. As such, the statement “Other groups of patients at risk for severe RSV disease also need vaccination: younger adults and children with immunocompromise, current tobacco smoking, diabetes, lung disease (e.g., asthma, chronic obstructive pulmonary disease), heart disease (e.g., congestive heart failure), liver disease, or renal disease” is changed to “RSV vaccination should be further studied in other patient groups at risk for severe RSV disease, e.g., younger adults and children with immunocompromise, and chronic cardiopulmonary disease.”
Line 343: Please change “renal disease” to “renal disease patients”.
Reply: This has been removed after the change above.
Line 344: The age limit of 18-64 years is unclear to me. Please justify or modify.
Reply: This has been removed.
Line 367: Please change “modified vaccinia Ankara” to “Modified Vaccinia Ankara (MVA)”.
Reply: Change made.
10. Conclusions
The smoothness of this paragraph gives me the impression that at least parts of it may have been written by AI. Please rephrase this section and list all AI tools used for the rest of the manuscript, if any.
Reply: No part of the manuscript was written by AI. I wrote the entire draft and used the built-in Editor function of MS Word to improve the spelling, grammar, and language of the entire manuscript. Nonetheless, I re-read the conclusions, and improved the smoothness by including a more active voice.
Line 404: Please remove “In conclusion”.
Reply: Word removed.
Line 413: Please remove “)”.
Reply: Removed.
Table 1
Column 1: Please add more information on the functional mechanism of the vaccines, e.g., bivalent recombinant protein subunit vaccine.
Reply: Added.
Please add a column on the regulatory approval by the European Medical Agency (EMA) and a column with references to the FDA and EMA approvals, respectively.
Reply: Added.
Please add the abbreviations RSV, COVID-19, Tdap, GSK, and U.S. to the subscripts of the table.
Reply: Added.
Table 2
Please add the abbreviations RSV, GSK, and U.S. to the subscripts of the table.
Reply: Added.
Table 3
This table is very bulky. However, it cannot entirely capture the complex information on each study (e.g, study design, primary and secondary outcomes, etc). Please consider extending the table by adding relevant columns or shortening it. Please sort by year of publication (descending).
Reply: Added a column for trial arms. Table entries sorted by vaccine type and then by year of publication, with sorting order mentioned in the footnote.
Please change “modified vaccinia Ankara” to “Modified Vaccinia Ankara (MVA)”.
Reply: Added.
Please add the abbreviations GSK, NB, and J&J to the subscripts of the table.
Reply: Added.
Line 329: Please change “Bacillus Calmette-Guerin” to “Bacillus Calmette–Guérin”.
Reply: Change made.
Table 4
This table is very bulky. However, it cannot entirely capture the complex information on each study (e.g, study design, primary and secondary outcomes, etc). Please consider extending the table by adding relevant columns or shortening it. Please sort by year of publication (descending).
Reply: Added a column for trial arms. Table entries sorted by vaccine type and then by year of publication, with sorting order mentioned in the footnote.
Please add the abbreviations GSK, NB, and J&J to the subscripts of the table.
Reply: Added.
Table 5
This table adds very little additional information for the reader. I recommend removing it.
Reply: Table 5 removed.
Table 6
This table is very bulky. However, it cannot entirely capture the complex information on each study (e.g, study design, primary and secondary outcomes, etc). Please consider extending the table by adding relevant columns or shortening it. Please sort by year of publication (descending).
Reply: Added a column for trial arms. Table entries sorted by vaccine type and then by year of publication, with sorting order mentioned in the footnote.
Please add the abbreviations GSK, and J&J to the subscripts of the table.
Reply: Added.
Line 359: Please change “Haemophilus influenzae” to “haemophilus influenzae” (in italic).
Reply: Species name italicized.
Reviewer 3 Report
Comments and Suggestions for Authors
This manuscript by Kay Choong See summarizes rich information on RSV, especially the treatment and vaccination aspects. The manuscript is well-written, scientifically sound, thorough, informational, and up-to-date. It is a well-polished manuscript and my only suggestion is to italicize species names such as Streptococcus pneumoniae in line 67
Author Response
Reviewer 3
This manuscript by Kay Choong See summarizes rich information on RSV, especially the treatment and vaccination aspects. The manuscript is well-written, scientifically sound, thorough, informational, and up-to-date. It is a well-polished manuscript and my only suggestion is to italicize species names such as Streptococcus pneumoniae in line 67.
Reply: Species name italicized.
Reviewer 4 Report
Comments and Suggestions for Authors
This review highlights essential aspects of RSV infection, treatment, and vaccination. The author made a good summary of the current vaccines available and in clinical studies. A few changes need to be performed before being accepted for publication.
Line 8: Respiratory syncytial virus.
The author needs to discuss extensively the current knowledge about extrapulmonary manifestations.
The RSV infection in pregnant women must be discussed extensively.
Line 135: RNA.
The cost-effectiveness of monoclonal use must be discussed.
Include references in Tables 1 and 5.
Avoid the excessive use of parentheses.
Review the redaction of the text because several parts of it are wordy.
Discuss how treatment may prevent extrapulmonary manifestations.
Comments on the Quality of English Language
Minor editing of the English language is required.
Author Response
Reviewer 4
This review highlights essential aspects of RSV infection, treatment, and vaccination. The author made a good summary of the current vaccines available and in clinical studies. A few changes need to be performed before being accepted for publication.
Line 8: Respiratory syncytial virus.
Reply: Capitalization removed from the original.
The author needs to discuss extensively the current knowledge about extrapulmonary manifestations.
Reply: Added a new paragraph on extrapulmonary manifestations of severe RSV disease in the section on “Clinical course of RSV infection”.
The RSV infection in pregnant women must be discussed extensively.
Reply: As it is unclear if adult vaccination for pregnant women is beneficial, I have included the following statement in the section “Future directions”: “Given that RSV-associated acute respiratory infections affect about 3.4% of pregnant women, with increased odds of preterm delivery, it will also be important to know if adult vaccination of pregnant mothers can benefit pregnancy outcomes.”
Line 135: RNA.
Reply: Original “DNA” changed to “RNA”.
The cost-effectiveness of monoclonal use must be discussed.
Reply: Immunoprophylaxis with monoclonal antibodies is cost-effective in both high income and resource-limited countries. This statement, backed with references, has been added to the section “General principles for prevention of RSV infection including vaccination”.
Include references in Tables 1 and 5.
Reply: References included in Table 1. Note that the original Table 5 has been removed following the recommendation of another reviewer.
Avoid the excessive use of parentheses.
Reply: I have reduced the number of parentheses.
Review the redaction of the text because several parts of it are wordy.
Reply: I have reduced parts of the text that are less important to the review aims.
Discuss how treatment may prevent extrapulmonary manifestations.
Reply: As no specific treatment for RSV exists, it remains unclear if treatment may prevent extrapulmonary manifestations. With regards to vaccination, prevention of severe RSV outcomes has not been well-studied. Therefore, I have included a suggestion to study the latter in the section on “Future directions”.
Round 2
Reviewer 1 Report
Comments and Suggestions for Authors
The authors accepted and reviewed all my recommendations, which they included in the manuscript. They also made some changes to the text improving its quality.